# Antiviral, Antibacterial, Antifungal, and Cytotoxic Silver(I) BioMOF Assembled from 1,3,5-Triaza-7-Phoshaadamantane and Pyromellitic Acid

**DOI:** 10.3390/molecules25092119

**Published:** 2020-05-01

**Authors:** Sabina W. Jaros, Jarosław Król, Barbara Bażanów, Dominik Poradowski, Aleksander Chrószcz, Dmytro S. Nesterov, Alexander M. Kirillov, Piotr Smoleński

**Affiliations:** 1Faculty of Chemistry, University of Wroclaw, F. Joliot-Curie 14, 50-383 Wrocław, Poland; sabina.jaros@chem.uni.wroc.pl; 2Department of Veterinary Microbiology, Wrocław University of Environmental and Life Sciences, Norwida 31, 50-375 Wrocław, Poland; jaroslaw.krol@upwr.edu.pl (J.K.); barbara.bazanow@upwr.edu.pl (B.B.); 3Department of Animal Physiology and Biostructure, Wrocław University of Environmental and Life Sciences, Kożuchowska 1, 51-631 Wrocław, Poland; dominik.poradowski@upwr.edu.pl (D.P.); aleksander.chroszcz@upwr.edu.pl (A.C.); 4Centro de Química Estrutural and Departamento de Engenharia Química, Instituto Superior Técnico, Universidade de Lisboa, Av. Rovisco Pais, 1049–001 Lisbon, Portugal; dmytro.nesterov@tecnico.ulisboa.pt; 5Research Institute of Chemistry, Peoples’ Friendship University of Russia (RUDN University), 6 Miklukho-Maklaya st., 117198 Moscow, Russia

**Keywords:** metal-organic frameworks, coordination polymers, silver complexes, 1,3,5-triaza-7-phosphaadamantane, pyromellitic acid, antimicrobial activity, cytotoxic activity, antiviral activity

## Abstract

The present study reports the synthesis, characterization, and crystal structure of a novel bioactive metal-organic framework, [Ag_4_(*µ*-PTA)_2_(*µ*_3_-PTA)_2_(*µ*_4_-pma)(H_2_O)_2_]_n_·6nH_2_O (bioMOF **1**), which was assembled from silver(I) oxide, 1,3,5-triaza-7-phosphaadamantane (PTA), and pyromellitic acid (H_4_pma). This product was isolated as a stable microcrystalline solid and characterized by standard methods, including elemental analysis, ^1^H and ^31^P{^1^H} NMR and FTIR spectroscopy, and single crystal X-ray diffraction. The crystal structure of **1** disclosed a very complex ribbon-pillared 3D metal-organic framework driven by three different types of bridging ligands (*µ*-PTA, *µ*_3_-PTA, and *µ*_4_-pma^4−^). Various bioactivity characteristics of bioMOF **1** were investigated, revealing that this compound acts as a potent antimicrobial against pathogenic strains of standard Gram-negative (*Escherichia coli*, *Pseudomonas aeruginosa*) and Gram-positive (*Staphylococcus aureus*) bacteria, as well as a yeast (*Candida albicans*). Further, **1** showed significant antiviral activity against human adenovirus 36 (HAdV-36). Finally, bioMOF **1** revealed high cytotoxicity toward an abnormal epithelioid cervix carcinoma (HeLa) cell line with low toxicity toward a normal human dermal fibroblast (NHDF) cell line. This study not only broadens the family of PTA-based coordination polymers but also highlights their promising multifaceted bioactivity.

## 1. Introduction

Infectious diseases represent the biggest healthcare concern worldwide [1]. This can be attributed to the observed decrease in the effectiveness of available antibacterial, antiviral, and antifungal agents and the lack of specific disinfectants [1,2,3]. Therefore, the search for new molecules, formulations and materials with potential bactericidal or antiviral properties is currently an extremely important research area. In contrast to numerous organic molecules with potent bioactivity, metal–organic compounds are less explored in this regard [1]. However, the combination of bioactive metals (e.g., Ag, Cu, Zn, Fe) along with pharmacologically active ligands or building blocks within one molecule can lead to compounds with particularly interesting properties.

In particular, bioactive metal–organic frameworks (bioMOFs) or coordination polymers emerged as a new class of compounds possessing almost boundless versatility in terms of structural types and property diversity with relevance to applications in many biomedical fields [3,4]. This is explained by high porosity, structural tunability, proper stability, versatile host–guest interactions, sorption, and ion release properties, along with many other features of such metal–organic architectures [3,5,6,7].

Among the metal ions with proven antibacterial and antifungal functions, silver represents the most interesting example, with its antimicrobial properties known for millennia [8,9]. Over the last two decades, a new range of applications for silver compounds was discovered, including the design of silver(I) bioMOFs with a variety of bioactive properties. These largely depend on the bond strengths between silver centers and ligand donor atoms, the Ag^+^ ion/ligand release behavior [8,9,10,11,12,13,14,15,16,17,18,19,20,21,22], and the solubility and stability of such compounds in aqueous medium. In turn, the bioavailability and solubility of coordination compounds in water or physiological media can be achieved by introducing ligands of appropriate lipophilicity to the coordination spheres of metal centers. To this end, the water-soluble and air-stable cagelike aminophosphine, 1,3,5-triaza-7-phosphaadamantane (PTA), was the ligand of choice when designing water-soluble metal complexes [23,24,25,26,27,28] and, more recently, bioactive coordination polymers [17,18,19,20].

Following our general research interest on the use of 1,3,5-triaza-7-phosphaadamantane as a versatile *P,N*-linker for the assembly of novel metal–organic architectures with diverse functional properties [16,17,18,19,20,21,22], in this work, we focused on further exploring a multicomponent system composed of silver(I) ions, PTA, and carboxylic acid building blocks to generate a new bioMOF. We report herein the synthesis, characterization, crystal structure, and a diversity of bioactivity tests for a novel 3D bioMOF [Ag_4_(*µ*-PTA)_2_(*µ*_3_-PTA)_2_(*µ*_4_-pma)(H_2_O)_2_]_n_·6nH_2_O (**1**) (H_4_pma = pyromellitic acid). This compound features a notable multifaceted bioactivity ranging from antibacterial and antifungal behavior to antiviral and cytotoxic properties, as described below.

## 2. Results and Discussion

### 2.1. Synthetic Procedure and Characterization

BioMOF Ag_4_(*µ*-PTA)_2_(*µ*_3_-PTA)_2_(*µ*_4_-pma)(H_2_O)_2_]_n_·6nH_2_O (**1**) was prepared using a conventional wet solvent synthesis with a multicomponent Ag_2_O−PTA−H_4_pma−NH_3_·H_2_O mixture of reagents in CH_3_OH/H_2_O, followed by slow crystallization (Scheme 1). As a result, **1** was isolated as a microcrystalline white solid and characterized by single-crystal X-ray analysis, elemental analysis, and standard FTIR and NMR spectroscopic methods.

A set of characteristic vibrations for the PTA and *µ*_4_-pma^4−^ ligands was observed in the FTIR spectrum of **1**. Bands in the 1200–900 cm^−1^ range are typical for coordinated PTA moieties [17,18,19,20,21,22]. Two sets of intensive *ν_as_*(COO) and *ν_s_*(COO) vibrations of *µ*_4_-pma^4−^ were observed at 1576–1569 cm^−1^ and 1486–1373 cm^−1^, respectively. The large Δ*ν* values (calculated frequency difference, Δ*ν* = *ν_as_*(COO) – *ν_s_*(COO) indicated that the monodentate coordination modes of all carboxylate groups in *µ*_4_-pma^4−^ were in good agreement with structural data [29,30]. Further, the characteristic *ν*(H_2_O) and *δ*(H_2_O) bands were detected at 3401 and 1670 cm^−1^, respectively. The ^1^H-NMR spectrum of **1** showed a set of signals expected for the H_3,6_ protons of µ_4_-pma^4−^ [18] along with the CH_2_ protons of PTA moieties. The ^31^P{^1^H} NMR spectrum disclosed a broad singlet at *δ* –78.2 that was consistent with the presence of Ag-PTA motifs in solution [17,18,19,20,21,22].

### 2.2. Crystal Structure 

The crystal structure of **1** (Figure 1) comprised four silver(I) atoms (two Ag1 and two Ag2), four PTA ligands (two pairs of *µ*- and *µ*_3_-bridging), a *µ*_4_-pma^4−^ pillar, two terminal H_2_O ligands, and six water molecules of crystallization per formula unit. The Ag1 centers were 4-coordinate and exhibited a distorted {AgPNO_2_} tetrahedral environment occupied by a P donor from *µ*-PTA, an N donor from *µ*_3_-PTA, a carboxylate O site of *µ*_4_-pma^4−^, and a terminal water ligand. The Ag2 atoms were also 4-coordinate and showed a distorted {AgPN_2_O} geometry, which was taken up by a P site of *µ*_3_-PTA, two aminophosphine N donors from the *µ*- and *µ*_3_-PTA moieties, and an O donor from *µ*_4_-pma^4−^. The Ag−P [2.335(3)–2.346(3) Å], Ag−N [2.335(3)–2.442(7) Å], and Ag−O_pma_ [2.253(6)–2.283(6) Å] bond distances were within typical values for Ag coordination polymers bearing PTA and carboxylate blocks [18,19,20]. The Ag1 and Ag2 centers were joined together via alternating *µ*- and *µ*_3_-PTA blocks resulting in 1D chain motifs (ribbons) composed of honeycomb-like [Ag_3_(PTA)_3_] subunits (Figure 1b). These 1D ribbons were interconnected by the *µ*_4_-pma^4−^ pillars (all COO^−^ groups are monodentate) to generate an intricate ribbon-pillared 3D metal-organic framework (Figure 1c). This framework was further reinforced by the O−H…O and O−H…N hydrogen bonds involving crystallization water molecules. Interestingly, there (H_2_O)_3_ clusters of the *D*3 type were formed via H-bonding of two crystallization water molecules and an H_2_O ligand [31,32,33].

The 3D MOF structure was also analyzed from a topological perspective [34,35], namely by generating a simplified topological net (Figure 1d). It was built from the 3- and 4-connected Ag1 and Ag2 nodes, the 3-connected *µ*_3_-PTA and 4-connected *µ*_4_-pma^4−^ nodes, and the 2-connected *µ*-PTA linkers. As a result, a tetranodal 3,3,4,4-linked net with unique topology was identified, described by the (5.7^2^)_2_(5^2^.7)_2_(5^2^.7^2^.8^2^)_2_(7^2^.8^2^.10^2^) point symbol with the (5.7^2^), (5^2^.7), (5^2^.7^2^.8^2^), and (7^2^.8^2^.10^2^) indices referring to the Ag1, *µ*_3_-PTA, Ag2, and *µ*_4_-pma^4−^ nodes, respectively.

### 2.3. Antibacterial and Antifungal Properties

The antimicrobial activity of the new compound **1** was assessed against three bacterial species (*Staphylococcus aureus, Escherichia coli,* and *Pseudomonas aeruginosa*) as well as the yeast *Candida albicans*. These microorganisms belong to important human and animal pathogens and often serve as representatives of the main groups of infectious agents, i.e., Gram-positive bacteria (*S. aureus*), Gram-negative bacteria (*E. coli and P. aeruginosa*), and fungi (*C. albicans*). Additionally, for all these pathogenic microorganisms, a marked increase in resistance to the commonly used drugs was noted [5,36,37]. Silver(I) bioMOFs, due to their potent antimicrobial properties and low toxicity, may constitute a promising alternative to conventional treatment protocols [9]. In the present work, we studied the antimicrobial activity of **1** against the above-listed microorganisms using silver nitrate as a reference Ag-containing antimicrobial. Efficiency was determined by the serial dilution method and expressed as the minimum inhibitory concentration (MIC) and normalized MIC (Table 1). The highest antibacterial activity of **1** was detected in the case of the Gram-negative bacteria, with the MIC values amounting to 5 μg·mL^−1^. Considering the silver content in the bioMOF **1**, significantly greater antibacterial efficacy was shown against Gram-negative bacteria than for silver nitrate. Generally, the normalized MIC (14 nmol·mL^−1^) was one third of that observed for AgNO_3_ (53 nmol·mL^−1^). Compound **1** was also active against *S. aureus* with the normalized MIC (22 nmol·mL^−1^) being five-fold lower than that exhibited by the reference silver salt (118 nmol·mL^−1^). A slightly different activity of **1** against the two main types of bacteria (Gram-negative and Gram-positive) may depend on the differences in their cell walls. A commonly accepted hypothesis suggests that Gram-negative bacteria, with a much thinner layer of peptidoglycan in the cell wall, allow silver ions to penetrate much better into the cells [1]. When compared to some other silver-conjugated MOFs described previously [14,15], compound **1** displayed similar activity against the Gram-negative bacteria and a distinctly stronger effect against *S. aureus*.

The new Ag(I)-bioMOF also revealed pronounced antifungal activity against *C. albicans*. The MIC value detected for this yeast (30 μg·mL^−1^) was unsurprisingly much higher than those observed for the bacteria. A similar situation was also noted for related MOFs [14,15]. The higher resistance of *C. albicans* may be attributable to the fact that fungi have relatively thick cell walls; however, some other mechanisms hampering the penetration of silver(I) species probably also play an important role. Nevertheless, the normalized MIC value for *C. albicans* (83 nmol·mL^−1^) was more than two times lower than that of the reference salt. Thus, the results of the antifungal activity for **1** reveal a new direction toward the development of topical fungicidal formulations.

### 2.4. Antiviral Activity 

Viral infections pose major public health risks, which are especially visible in present-day epidemics such as COVID-19 in humans or African Swine Fever (ASF) in animals, thus justifying intensive research into developing new antiviral molecules and materials. Silver compounds might also be suitable for this purpose, as attested by prior research. For example, Shimizu et al. showed activity of AgNO_3_ against herpes simplex virus type 1 (HSV-1) [39], while silver nanoparticles were effective in destroying HIV, hepatitis B, herpes simplex, respiratory syncytial, and monkey pox viruses [40]. Silver ions from silver nitrate and silver(I) oxide also exhibited strong inactivation of influenza A virus (A/PR8/H1N1) [41].

Human adenovirus 36 (HAdV-36), which was used in the present work, is a non-enveloped lytic DNA virus with a linear double-stranded genome and icosahedral symmetry [42]. This virus causes respiratory and eye infections, as well as obesity in humans and animals. In seroepidemiological surveys in people, the prevalence of HAdV-36 increases in relation to body mass index. Therefore, infection by HAdV-36 should be considered as a possible risk factor for obesity and could be a potential new therapeutic target, in addition to an original way to understand the worldwide rise of the epidemic of obesity [43]. Most of the reported compounds with anti-adenovirus activity are nucleoside or nucleotide analogues, such as cidofovir [(*S*)-HPMPC; (*S*)-1-(3-hydroxy-2-phosphonylmethoxypropyl)cytosine], (*S*)-HPMPA [(*S*)-9-(3-hydroxy-2-phosphonylmethoxypropyl)-adenine] and 2-nor-cyclic GMP {9-[(2-hydroxy-1,3,2-dioxaphosphorinan-5-yl)oxymethyl]-guanine phosphate oxide}, with a number of side effects [42]. While there were also reports regarding other substances active against adenoviruses, including sulfated sialyl lipid NMSO3 {sodium [2,2-bis(docosyl-oxymethyl)propyl-5-acetoamido-3,5-dideoxyl-4,7,8,9-tetra-*O*-(sodium-oxy sulfonyl)-d-glycero-d-galacto-2-nonulopyranosid]onate} or the endogenous microbicide *N*-chlorotaurine [44], silver compounds remain poorly studied. Silver nanoparticles on a magnetic hybrid colloid (AgNP-MHC) were efficient against bacteriophage Φ X174 and murine noroviruses (MNV), but not against adenovirus AdV2 [45]. So far, only Chen et al. were able to demonstrate a remarkable inhibitory effect of silver nanoparticles on adenovirus [46].

According to EN 14476:2005 [47], a disinfectant is considered as having virucidal effectiveness if the titer is reduced by ≥4 log_10_ steps within the recommended exposure time (inactivation of ≥99.99%). Compound **1** was examined in a below-toxicity concentration (50 μM) to normal human cell dermal fibroblast (NHDF) cells within 30 min exposure time. After this time, reduction factors reached a value of ≥4.00 log_10_, making **1** a promising therapeutic or disinfecting agent.

### 2.5. Cytotoxic Properties on Normal and Cancer Cell Lines

Three cell lines were used in this study. Two (HeLa and A549) belong to one type of neoplasm derived from epithelial tissue. This type of neoplasm (carcinoma) has a high potential for metastasis and hardly responds to routine anticancer treatment. As a normal cell body reflection, normal human dermal fibroblasts (NHDF) were used. Similar utilization of this NHDF cell line can be found in the literature [48,49]. A cytotoxicity assay was carried out using the MTT (3-(4,5-dimethylthiazol-2-yl)-2,5-diphenyltetrazolium bromide) test. The half maximal inhibitory concentration (IC_50_) values after 72 h of incubation with bioMOF **1** are shown in Table 2.

The obtained results proved that bioMOF **1** was cytotoxic against the normal human dermal fibroblasts cell line (NHDF) and the human cervix carcinoma cell line (HeLa). However, there was no cytotoxicity against the human lung carcinoma cell line (A549). For the NHDF cell line, the cytotoxicity of **1** was comparable to silver nitrate. For the HeLa cell line, the cytotoxicity of **1** was significantly higher (IC_50_ 33.0 ± 4.1 µM) than that of AgNO_3_ (IC_50_ 176.6 ± 33.2). The difference was even more pronounced if the observed cytotoxicity was normalized for the molar content of silver in **1** and AgNO_3_. This was an important observation, because the change in the chemical structure increased the cytotoxicity to neoplastic cells, while maintaining a similar degree of cytotoxicity effect to normal body cells. This fact may be of crucial clinical importance, because it may open up potential for **1** as a promising anticancer drug after additional in vitro and in vivo studies. Looking at the absence of cytotoxic activity of **1** and the weak effect of silver nitrate on the A549 line, these observations may be explained by a previously developed resistance of cells to AgNO_3_. Furthermore, both free ligands, H_4_pma and PTA, showed no cytotoxic effects on any of the three cell lines tested. The absence of cytotoxicity of both ligands seems to be beneficial when used in practice, specifically by eliminating the risk of possible side effects that may be observed in vivo. While comparing the degree of cytotoxicity of **1** and AgNO_3_ versus a routinely used anticancer drug, cisplatin, the former two show weaker cytotoxic effects. However, the results obtained for **1**, especially in relation to cervical cancer cells, are very promising. It is also worth emphasizing that **1** is more soluble in aqueous solutions than cisplatin, which may also be an advantage of the tested bioMOF. However, despite being outside of the scope of the present work, further biological and clinical studies should be performed to better understand the possible advantages and limitations of bioMOF **1**.

## 3. Experimental

### 3.1. Materials and Methods

The materials, chemicals, and solvents were purchased from commercial sources and used as received without further purification, except PTA (1,3,5-triaza-7-phosphaadamantane), which was synthesized according to a published method [52,53]. C, H, and N elemental analyses were carried out using the Elemental Analyser VarioELCube (Elementar Analysen systeme GmbH, Hanau, Germany) by the Laboratory of Elemental Analysis at Faculty of Chemistry, University of Wrocław. Infrared (FTIR) spectra were performed on a Bruker IFS 1113v (Ettlingen, Germany) or BIO-RAD FTS 3000MX (BIO-RAD, Paris, France) instrument in the 4000–400 cm^−1^ range by the Laboratory of Infrared Spectroscopy at the Faculty of Chemistry, University of Wrocław (abbreviations: vs, very strong; s, strong; m, medium; w, weak; br., broad). ^1^H and ^31^P{^1^H} NMR spectra were measured using a Bruker 500 AMX spectrometer (Bruker BioSpin MRI GmbH, Ettlingen, Germany,) at ambient temperature (abbreviations: s, singlet; d, doublet; br., broad).

### 3.2. Antibacterial and Antifungal Activity Studies

The antimicrobial activity of **1** and AgNO_3_ (used as a reference) were evaluated by the broth dilution method, as described previously [19,38,54,55,56,57]. Briefly, serial dilutions of the compound in antibiotic broth (AB) of 0.9 mL each were prepared in 48-well microtiter plates. Next, 0.1 mL of an overnight bacterial or fungal culture in AB, diluted to 1:1000, was added to each well. The final concentrations of the substance were as follows: 60, 50, 40, 30, 20, 10, 9, 8, 7, 6, 5, 4, 3, 2, and 1 µg·mL^−1^. The microorganisms tested were two reference strains, i.e., *Staphylococcus aureus* (ATCC 25923) and *Escherichia coli* (ATCC 25922), both obtained from the Polish Collection of Microorganisms, and two clinical isolates, i.e., *Pseudomonas aeruginosa* and *Candida albicans*. The latter two strains were isolated from veterinary samples and identified by means of routine microbiological methods. Inoculated plates were incubated at 37 °C for 24 h and then checked for microbial growth (manifested as visible cloudiness of the medium). The lowest concentration of a substance that completely inhibited the growth of a given microorganism was reported as minimum inhibitory concentration (MIC). Additionally, a normalized MIC was calculated to express the antimicrobial efficacy in relation to the molar content of silver in **1** versus AgNO_3_ (nmol·mL^−1^).

### 3.3. Cell Cultures

Cytotoxicity evaluations were carried out using the cell cultures of normal human dermal fibroblasts (NHDF; PromoCell, C-12302^®^, Heidelberg, Germany), human lung carcinoma (A549; ATTC, No CCL-185 ^®^), and human cervix carcinoma (HeLa; ATCC, No CCL-2 ^®^). The two first cell lines were cultured in Dulbecco’s Modified Eagle Medium (DMEM, Lonza, Basel, Switzerland) and the third one was cultured in Minimum Essential Medium (MEM (Sigma, Steinheim, North Rhine-Westphalia, Germany). All media were supplemented with 10% Fetal bovine serum (FBS, Biological Industries, Kibbutz Beit-Haemek, Israel), 4 mM L-glutamine (Biological Industries, Kibbutz Beit-Haemek, Israel), 100 U/mL of penicillin, and 100 μg/mL of streptomycin (Sigma, Steinheim, North Rhine-Westphalia, Germany).

### 3.4. Cytotoxicity Assay

During the study, 96-well plates (Eppendorf, Freie und Hansestadt Hamburg, Germany) were used for cell culture insertion in a concentration of 10^5^ cells per well. **1** was dissolved in culture medium and diluted to afford concentrations of 100, 50, 20, 10, 5, 2, and 1 µg·mL^−1^. The plates were incubated in standard conditions (37 °C with a constant flow of 5% CO_2_) for 72 h. Then, the MTT assay (Sigma, Steinheim, North Rhine-Westphalia, Germany) was used to evaluate the cytotoxicity proprieties of mentioned compounds. The MTT procedure consisted in adding 20 μL of MTT (mg·mL^−1^) to each well and subsequently incubating for 4 h at 37 °C. After incubation, the lysis buffer (80 μL) was added in order to destroy the cell membrane. The basis of the MTT assay is enzymatic reduction of soluble tetrazolium salt in metabolically active cells into insoluble purple formazan. The concentration of formazan was measured colorimetrically using a spectrophotometric microplate reader (Multiscan Go, Thermo Fisher, Waltham, MA, USA). As a result of the spectrophotometric analysis, the optical density (OD) was computed. The viability of the investigated cell cultures was estimated using the following formula: viability% = (average OD for test group/average OD for control group) × 100. The untreated cells were used as a control group.

### 3.5. Virucidal Activity According to PN-EN 14476

Adenocarcinomic human alveolar basal epithelial cells (A549) (ATTC, Manassas, VA, USA, No CCL-37TM) were incubated in 96-well polystyrene plate for 24 h for use in this project. Human adenovirus 36 virus (HAdV-36, ATCC^®^ VR-1610™) was used to check the virucidal activity of bioMOF **1**.

### 3.6. Antiviral Assay

In a typical procedure, 8 mL of **1** in a concentration of 50 µM was added to a suspension containing 1 mL of 100 TCID_50_ HAdV-36 virus and 1 mL of interfering substance (0.30 g of bovine albumin fraction V, suitable for microbiological purposes, in 100 mL of water). After mixing, the solutions were placed in a water bath controlled at 56 °C for 30 min. At the end of this contact time, aliquots were taken, with the virucidal action in these portions immediately suppressed by a validated method (dilution of the sample in ice-cold cell maintenance medium). Serial dilutions up to 10-8 of each mixture were prepared and 50 µL of each dilution in eight replications was transferred to the microtiter plate containing a monolayer of confluent A549 cells. The plate was observed daily for up to 7 days for the development of viral cytopathic effects using an inverted microscope (Olympus Corp., Hamburg Germany; Axio Observer, Carl Zeiss MicroImaging GmbH). Reduction of virus infectivity was calculated from differences of log virus titers before (virus control) and after treatment with the product.

### 3.7. Synthesis and Characterization of BioMOF ***1***

[Ag_4_(*µ*-PTA)_2_(*µ*_3_-PTA)_2_(*µ*_4_-pma)(H_2_O)_2_]_n_·6nH_2_O (**1**). Silver(I) oxide (0.1 mmol, 23 mg) was combined with pyromellitic acid (H_4_pma, 0.25 mmol, 63.5 mg) in a solvent mixture consisting of methanol (7 mL) and water (3 mL). The obtained suspension was stirred for 45 min in air at room temperature, then solid 1,3,5-Triaza-7-phoshaadamantane (PTA, 0.2 mmol, 31.4 mg) was introduced. The obtained white mixture was stirred for an additional 45 min, resulting in the formation of a white suspension. This was treated with a 1 M aqueous solution of NH_3_·H_2_O (ca. 1.0 mL) until complete dissolution (pH = 8.5) and then filtered off. The obtained filtrate was left in a vial for several days to slowly evaporate in air, leading to the formation of colorless crystals (including single crystals suitable for X-ray diffraction). These were washed with methanol, dried, and collected to produce **1** in 35% yield (based on silver(I) oxide). Anal. Calcd. for C_34_H_66_Ag_4_N_12_O_16_P_4_ (MW = 1454.34) C, 28.08, H, 4.57, N, 11.56; found for: C, 28.10, H, 4.51, N, 11.48; FTIR (KBr, cm^−1^): 3401 (br s), 3271 (w), 2957 (w) 2918 (w), 1670 (w), 1576 (vs), 1476 (w), 1449 (w), 1414 (w), 1363 (s), 1316 (w), 1283 (m), 1237 (m), 1133 (m), 1100 (m), 1037 (m), 1013 (s), 979 (w), 971 (vs), 951 (vs), 933 (w), 901 (w), 892 (w), 860 (m), 795 (s), 756 (m), 736 (w), 671 (w), 604 (m), 596 (m), 563 (m), 491 (w), 454 (m), 419 (w). ^1^H-NMR (500 MHz, D_2_O, Me_4_Si): *δ* 7.74 (s, 2H, H_3,6_, pma), 4.65 and 4.55 (2d, 24H, *J*_AB_ = 14.0 Hz, NCH_A_H_B_N, PTA), 4.26 (s br, 24H, PCH_2_N, PTA). ^31^P{^1^H} NMR (202.5 MHz, D_2_O, 85% H_3_PO_4_): *δ* −77.9 (s, PTA).

### 3.8. X-ray Crystallography

Crystal data for **1**: C_34_H_66_Ag_4_N_12_O_16_P_4_, M = 1454.34, *a* = 11.6333(7) Å, *b* = 11.8739(13) Å, *c* = 18.0211(11) Å, *β* = 98.011(5)°, *V* = 2465.0(3) Å^3^, *T* = 100.01(10) K, space group *P*2_1_/c, *Z* = 2, Mo*K*α, 11,345 reflections measured, 5450 independent reflections (*R*_int_ = 0.0645). The final *R*_1_ value was 0.0725 (*I* > 2σ(I)). The final *wR*(*F*^2^) value was 0.1766. The goodness of fit for *F*^2^ was 1.177.

Single-crystal data collection was performed using a KUMA diffractometer with a Sapphire CCD detector equipped with an Oxford Cryosystems open-flow nitrogen cryostat, using ω-scan and a graphite-monochromated Mo K (λ = 0.71073 Å) radiation. Cell refinement, data reduction, analysis, and absorption correction were carried out using CrysAlis PRO (Rigaku Oxford Diffraction, Wrocław, Poland) software [58]. The structure was solved by direct methods with SHELXT-2014/5 and refined with the full-matrix least-squares techniques for *F*^2^ with SHELXL-2018/3 [59,60]. The unusually low *U*_eq_ of N6 suggested that its site was partially occupied by an atom heavier than nitrogen. Thus, the respective PTA ligand was modeled as being disordered, with the atoms P2 and N6 exhibiting mutual substitutional disorder and the site occupancies refined to 0.774(12) and 1–0.774(12). The hydrogen atoms of water molecules were located and refined by restraining the O–H bond lengths and H–O–H angles to ideal values. All other hydrogen atoms were placed at calculated positions and refined using the model *U*_iso_ = 1.2*U*_eq_. Structural visualization was prepared using Diamond [61] or Topos software (Samara, Russia) [34,35]. CCDC-1996731 contains the supplementary crystallographic data for this paper.

## 4. Conclusions

In this study, we further extended the limited family of silver(I)-based bioMOFs to a novel heteroleptic example [Ag_4_(*µ*-PTA)_2_(*µ*_3_-PTA)_2_(*µ*_4_-pma)(H_2_O)_2_]_n_·6nH_2_O (1) with notable antibacterial, antifungal, antiviral, and cytotoxic activities. This product was fully characterized, and its crystal structure revealed a very complex 3D metal–organic architecture constructed from honeycomb-like [Ag_3_(PTA)_3_]_n_ ribbons and pyromellitate(4−) pillars. This work also widened the restricted application of PTA as a water-soluble cagelike *P,N*-linker in designing 3D framework structures.

Different bioactivity properties of **1** were studied, disclosing its high potential to inhibit the growth of pathogenic Gram-negative (*E. coli*, *P. aeruginosa*) and Gram-positive (*S. aureus*) bacteria, as well as yeast (*C. albicans*), with the normalized MIC (minimum inhibitory concentrations) values significantly lower than those of the conventional topical silver(I) antimicrobial (AgNO_3_). Apart from bacteria and fungi, bioMOF **1** also exhibited significant antiviral activity against HAdV-36, as well as high cytotoxicity toward the abnormal epithelioid cervix carcinoma (HeLa) cell line. The obtained results regarding such a multifaceted bioactivity of **1** deserve further exploration, which is currently in progress.

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
