# Peer review of "Antiviral, Antibacterial, Antifungal, and Cytotoxic Silver(I) BioMOF Assembled from 1,3,5-Triaza-7-Phoshaadamantane and Pyromellitic Acid"

_molecules, 2020, doi:10.3390/molecules25092119_

Round 1

Reviewer 1 Report

The paper reports a new Ag(I)-bioMOF, its synthesis and chemical characterization, and an investigation on its bioactivity. In particular, the structural analysis evidences a fascinating arrangement of the molecule. On the other hand, the study of the bioactivity allows widening the knowledge on the antimicrobial activity of Ag(I) derivatives.

The manuscript is well written and clearly presented and discussed. I recommend accepting this paper with minor revisions, as detailed in the following comments.

1) Scheme 1. Please, correct the text under the arrow (it should be NH3.H2O, but some overlapping in letters happened).

2) Table 2 and text (line 203). The value of the standard deviation is expressed with an excess of digits: strictly speaking, 2.36 and 4.06 as standard deviations do not have statistical significance. The correct form should be 2 and 4, respectively, and the final values could be "50.5±2" (being 5 the first decimal) and "33±4". Anyway, given the expressions in literature data here reported, I could convince myself to accept 50.5±2.4 and 33.0±4.1, respectively.

3) Lines 265 and 266. I suppose that correct terms are "soluble" and "insoluble", instead of "solube" and "insolube", respectively.

4) Line 324. A bracket is closed but not opened.

5) Lines 333, 337, 339. "1" is in plain character, instead of in bold as in all of the paper. 

Author Response

Cover Letter / Response to Reviewers

Dear Editor,

We would like to submit for your consideration a revised version of our manuscript (ID: molecules-790031) that has been amended taking into consideration the points indicated by the Reviewers 1 and 2. We thank the Reviewers for identifying a number of important points that allowed us to significantly improve the quality of the manuscript. Our point-by-point responses to reviewers’ comments are given below.

Reviewer: 1

We thank the Reviewer for a positive evaluation of our work and for the very valuable suggestions aiming at its further improvement.

Comment 1: Scheme 1. Please, correct the text under the arrow (it should be NH3.H2O, but some overlapping in letters happened).

Answer: Corrected.

Comment 2: Table 2 and text (line 203). The value of the standard deviation is expressed with an excess of digits: strictly speaking, 2.36 and 4.06 as standard deviations do not have statistical significance. The correct form should be 2 and 4, respectively, and the final values could be "50.5±2" (being 5 the first decimal) and "33±4". Anyway, given the expressions in literature data here reported, I could convince myself to accept 50.5±2.4 and 33.0±4.1, respectively.

Answer: Corrected.

Comment 3: Lines 265 and 266. I suppose that correct terms are "soluble" and "insoluble", instead of "solube" and "insolube", respectively.

Answer: Corrected.

Comment 4: Line 324. A bracket is closed but not opened.

Answer: Corrected.

Comment 5: Lines 333, 337, 339. "1" is in plain character, instead of in bold as in all of the paper.

Answer: Corrected.

Reviewer: 2

We thank the Reviewer for the positive evaluation of our work and for the very valuable suggestions aiming at its further improvement.

Comment 1: It would have been better if the antibacterial and antifungal activities of the new compound would have been evaluated using at least one systemic antibiotic agent as positive control, and an antifungal agent, respectively.

Answer: We agree with the Reviewer that adding some information regarding antimicrobial activity of selected antibacterial and antifungal drugs might be useful to highlight the effectiveness of the newly synthesized MOF. However, there is a number of such antimicrobials varying greatly in chemical composition, mode of action, inhibitory concentration window, and efficacy, so that the use of a single, randomly chosen antimicrobial as a control would be, in our opinion, less reliable than the application of a simple water-soluble salt of silver (silver nitrate), which is well-known and broadly referenced topical antibacterial that also possesses an antifungal activity (for example see References in manuscript 15-17, 19-22 and 38). Given the application of silver nitrate as primary pharma product to treat some topical bacterial and fungal infections, we believe the comparison of the activity of Ag bioMOF 1 with that of silver nitrate is the most appropriate.

In addition, given a current situation with the COVID-19 pandemic, we are currently very limited in performing any additional experimental work. Based on the above reasons, we believe the revised manuscript can be submitted in the present form. We thank you for your understanding.

Following your request, all changes in the manuscript are marked by a blue background.

We hope the manuscript is now in a suitable form to be considered for publication. Please do not hesitate to contact us if any other clarification or amendment is required.

We thank you for your kind consideration of our manuscript.

Looking forward to hearing from you,

With best regards,

                                                               Sincerely,

Piotr Smoleński

(on behalf of the authors)

Reviewer 2 Report

The paper makes an original and interesting contribution to the field of metal-organic frameworks with biological activity. The chemical part is well structured and the compound’s characterization is rigorously performed. It would have been better if the antibacterial and antifungal activities of the new compound would have been evaluated using at least one systemic antibiotic agent as positive control, and an antifungal agent, respectively.

Author Response

(The authors gave the same response as above.)
